# Relationships among Permeability, Membrane Roughness, and Eukaryote Inhabitation during Submerged Gravity-Driven Membrane (GDM) Filtration

**Dongwhi Lee [1], Yun Jeong Cha [2], Youngbin Baek [3], Shin Sik Choi [2,4],* and Yunho Lee [1],***

[1] School of Earth Sciences and Environmental Engineering, Gwangju Institute of Science and Technology (GIST), Gwangju 500-712, Korea; dongwhi7222@gist.ac.kr

[2] Department of Food and Nutrition, College of Natural Science, Myongji University, Yongin, Gyeonggi 449-728, Korea; yjcha2013@gmail.com

[3] Department of Biotechnology, Sungshin Women's University, Seoul 01133, Korea; ybbaek@sungshin.ac.kr

[4] Department of Energy Science and Technology, Myongji University, Yongin, Gyeonggi 449-728, Korea

\* Correspondence: sschoi@mju.ac.kr (S.S.C.); yhlee42@gist.ac.kr (Y.L.)

**Abstract:** Gravity-driven membrane (GDM) filtration is one of the promising technologies for decentralized water treatment systems due to its low cost, simple operation, and convenient maintenance. The objective of this study was to evaluate the permeability of submerged GDM filtration with three different membranes, i.e., polyethersulfone and polyvinylidene difluoride ultrafiltration (PES-UF and PVDF-UF) and polytetrafluoroethylene microfiltration membrane (PTFE-MF). The GDM system was operated using lake water for about one year. The determined average permeability values were high for PVDF-UF (192.9 L/m$^2$/h/bar (LMH/bar)) and PTFE-MF (80.6 LMH/bar) and relatively lower for PES-UF (46.1 LMH/bar). The observed higher permeability for PVDF-UF and PTFE-MF was thought to be related to the rougher surface of these two membranes compared to PES-UF. The fouling layers of PVDF-UF and PTFE-MF were characterized by high biomass and the presence of a number of nematodes, while PES-UF showed a thin fouling layer with no nematode. The relatively high and fluctuated permeability of PVDF-UF and PTFE-MF could thus be attributed to the high biological activity of nematodes making the fouling layer more loose and porous. This was supported by a good linear relationship among the permeability, biomass concentration, and the number of nematodes in the fouling layers. These results provide important insights into membrane selection as a critical factor affecting the flux performance of the GDM filtration system for a decentralized drinking water supply.

**Keywords:** gravity-driven membrane (GDM); permeability; surface roughness; nematode; biomass concentration; intermittent operation mode

## 1. Introduction

Contaminated water sources are becoming a tremendous problem in developing and transient countries [1]. Since the installation of a centralized drinking water treatment system requires a high energy input and a large footprint, a decentralized drinking water treatment system at the household or community level is an alternative option for a sustainable drinking water supply [2]. Gravity-driven membrane (GDM) filtration is one of the promising decentralized drinking water treatment systems due to its low cost, simple operation, and convenient maintenance [3–6].

The GDM filtration system usually operates in a dead-end mode at ultralow pressure (<0.1 bar), which leads to a relatively low but stable permeate flux (<10 L/m$^2$/h (LMH)) over several months [6]. Although membrane fouling phenomena decrease the filtration performance in water treatment systems [7], the ultralow transmembrane pressure in GDM filtration systems favors the formation of a loosely bound fouling layer on the membrane, which creates a pathway for the treated water. Wu et al. [4] found that over 250 days a thicker but porous fouling layer with less accumulation of organic substances improved the permeate flux in GDM filtration. Desmond et al. [8] reported that the composition of extracellular polymeric substances (EPS) from biofilm formation determined the physical structure of the fouling layer and eventually affected the hydraulic resistance during GDM filtration. The presence of a diverse microbial community, including eukaryotic organisms in the fouling layer, could affect flux behavior during GDM filtration. For example, the movement and predation activity of metazoans made the fouling layer more porous and heterogeneous, resulting in temporary flux recovery [9,10].

The surface properties of membranes can play an important role in membrane fouling phenomena [11,12]. It is generally known that membrane properties, including hydrophilicity and roughness, are closely related to membrane fouling; more hydrophobic and rougher surface membranes typically show decreased permeate fluxes due to facile attachment and accumulation of foulants on the membrane surface. In terms of the effect of surface charge on the membrane fouling phenomenon, negatively charged membranes usually exhibit fouling resistance, although some literature reported that positively charged membranes or zwitterionic membranes were more effective against membrane fouling [13]. In GDM filtration systems, a more hydrophilic membrane could increase permeate flux in the initial period of operation, but contributes less to the stabilized permeate flux in the extended operation period [14]. Thus, by using surface membrane modification, it has been tried to raise the low permeate flux in the GDM process to the level of conventional membrane filtration systems, whose permeate flux ranges from 50 to 100 LMH at 0.2–1.0 bar [15]. The modified membranes with amphiphilic multi-arms polymer, zwitterion polymer, and powdered activated carbon (PAC)/zeolite particles enhanced the hydrophilicity of the membrane surface, resulting in 50–70% increased permeate fluxes in the GDM system [16–18]. However, the effect of membrane surface properties on the eukaryotic community composition within the fouling layer was not investigated.

The objective of this study was to investigate the effects of surface properties on permeability behavior related to the nematode population, as a representative metazoan species, on the fouling layer. A submerged GDM filtration system with an intermittent operation mode was used to simulate the practical process for approximately 400 days (~100 cycles in GDM operation). Two ultrafiltration membranes and one microfiltration membrane with different surface properties were tested using lake water as the input. During long-term operation, the structure and characteristics of the membrane fouling layers were analyzed, and the number and identity of nematodes found in the membrane fouling layers were analyzed to obtain additional insights into the surface properties affecting the activity of nematodes and the flux performance of GDM filtration.

## 2. Materials and Methods

### 2.1. Membrane Characterizations

Two UF membranes and one MF membrane were used in this study. Polyethersulfone (PES, YMUE503001) and polyvinylidene difluoride (PVDF) UF membranes were purchased from Trisep Co. and Philos Co., Ltd. and the nominal molecular weight cut-off (MWCO) of the two UF membranes was 100 kDa. Polytetrafluoroethylene (PTFE) MF membrane, with a mean pore size of 0.3 μm, was purchased from AMTS Co., Republic of Korea. The membranes were stored in deionized (DI) water for 24 h to remove the membrane conservation agents and then placed in flat sheet membrane modules. The three membranes have the same effective membrane area of 520 cm$^2$ (40 cm × 13 cm).

The micro-scale structure of the membrane, such as morphology and roughness was characterized by using atomic force microscopy (AFM, PISA XE-100, Suwon, Korea) in non-contact mode. The membrane area for the AFM measurement was $45 \times 45$ $\mu m^2$. Images were scanned at a rate of 1 Hz. A mean roughness of membrane surface ($R_a$) was calculated as an average value of each scan line for the $10 \times 10$ micron images. The contact angle of the membranes was measured with the captive bubble method using a goniometer and the contact angle calculation was performed using the instrument software (DSA 100, Krüss, Hamburg, Germany) [19]. Before the contact angle measurement, the membranes were rinsed and dried in a closed desiccator for 24 h and stored in closed Petri dishes. Zeta potential of the clean membrane surface was measured using an electrophoretic light scattering spectrophotometer (ELS-8000, Ostuka Electronics, Osaka, Japan) [20–22]. Its value was determined from electrophoretic mobility measurements. Neutral polystyrene latex particles (diameter 520 nm, Otsuka Electronics, Osaka, Japan) were used for the measurement of mobility monitoring particles [23]. These were dispersed in a NaCl solution (1 mM) at pH 7.0 to prevent the interactions with, or adsorption on, the quartz cell surface. The water permeability of the clean membranes was evaluated using Amicon stirred cell (UFSC05001, Merck Millipore, Burlington, MA, USA) in dead-end mode with DI water at 1 bar.

## 2.2. Gravity Driven Membrane Filtration System

Figure 1 shows the schematic of the submerged GDM filtration system which was operated in a dead-end filtration in the intermittent operation mode. The GDM reactor was equipped with vertically installed membrane modules. The feed tank was initially filled with feed water at the height of 97.5 cm and the filtration carried out until the water level decreased to 32.5 cm. The remaining water was discarded and the tank was filled again to start the next cycle. This intermittent mode was used to simulate a practical GDM filtration operation [24–26]. The hydraulic pressure varied from 0.075 bar to 0.01 bar during the filtration (i.e., one cycle), calculated from the height of feed in the tank at 97.5 cm and 32.5 cm, respectively. The permeate line was located at a height of 22.5 cm. The GDM reactor (97.5 cm $\times$ 31 cm $\times$ 31 cm, 93 L of feed water volume) was operated without any back flushing and membrane cleaning for 99 cycles for PES-UF (or 354 days) and 116 cycles for PVDF-UF and PTFE-MF (or 416 days). Each cycle usually took 3–4 days. Note that this long-term experiment was interrupted a few times due to the lack of feed water supplement. The feed water tank was cleaned twice a month to minimize sedimentation and accumulation of particles on the tank bottom.

Lake water, which was taken from the influent to the drinking water treatment plant located in Gwangju (Korea), was used as the feed water. The feed water characteristics during the GDM operation were; dissolved organic carbon (DOC) of 2–4 mgC/L, dissolved oxygen (DO) of 6–12 mg/L, turbidity of 1.1–4.4 NTU, pH of 6.5–8.8, and temperature of $20 \pm 2$ °C. The permeate flux (L/m$^2$/h; LMH) was calculated by measuring the volume of permeate water collected in volumetric flasks within a certain filtration time (5–7800 s) and dividing by the effective membrane area (520 cm$^2$). The permeate flux was measured every operation day according to the change in the water level (from 0.075 bar to 0.01 bar). The mean permeability (LMH/bar) was used as an average value of permeate flux divided by hydraulic pressure during one cycle.

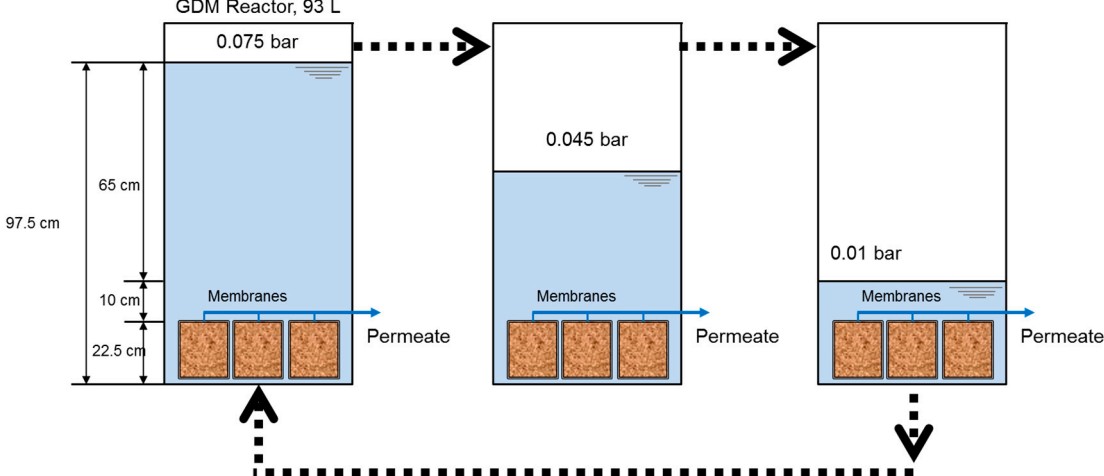

**Figure 1.** Experimental set-up of the submerged gravity-driven membrane (GDM) filtration system in the intermittent operation mode. Each cycle of the GDM operation consisted of feed water filling and discharging. The next cycle was started by emptying the remaining water and refilling with the feed water. One cycle indicates the changes in permeability according to the water level (from 0.075 bar to 0.01 bar corresponding to the height of 97.5 cm and 32.5 cm).

## 2.3. Analysis of Membrane Fouling Layer and Dissolved Organic Matter

The membrane surfaces during GDM filtration were periodically recorded using a digital camera (Canon EOD 100D) to obtain macroscopic observation of membrane fouling layers. To quantify the membrane coverage by the fouling layer, the 'Image J' software (http://rsb.info.nih.gov/ij/) was used. The automatic triangle algorithm was then applied to distinguish between the fouling layer and the uncovered membrane surface. The membrane coverage was obtained by calculating the selective surface area of the membrane [10,27]. The microscopic structure of the fouled and clean membrane was characterized using a Scanning Electron microscope (SEM, Hitachi S-4800, Tokyo, Japan) after the long-term operation. Sample preparation was done using the freeze-drying method (at low temperature ($-55$ °C) and vacuum for 24 h) to remove the water content and preserve the structure of the fouling layer. The samples were then coated with platinum using an ion sputter coater (E-1030, Hitachi, Tokyo, Japan). Top-view and cross-sectional SEM images were visualized at an accelerating voltage of 10 kV.

The biological activity of the fouling layer was determined by adenosine triphosphate (ATP) measurement using BacTiter-Glo reagent (Promega Corporation, Madison, WI, USA). A small fraction of the membrane fouling layers were carefully detached by a sterile cell scraper and transferred to DI water, and then homogenized. The samples of 500 µL and ATP reagent of 50 µL were warmed for 1 min at 38 °C in separate sterile tubes followed by combining them for the reaction for 20 s incubation time at 38 °C under dark conditions [28].

Size exclusion chromatography coupled to an organic carbon detector (SEC-OCD) was used to determine the chromatographically separable organic matter fraction according to molecular weight and charge. There were four fractions of raw water: (1) biopolymer (>10 kDa), (2) humic and building block (0.5–1 kDa), (3) low molecular organic acids (<0.5 kDa), and (4) low molecular neutral compounds (<0.5 kDa). The SEC-OCD includes high-performance liquid chromatography (Agilent 1260, Palo Alto, CA, USA) with a size exclusion column (TSK HW-50s, 3000 theoretical plates, Tosoh, Tokyo, Japan) and an organic carbon detector (Sievers Portable turbo total organic carbon analyzer 9000, Boulder, CO, USA) [29,30]. The online mobile phase (Phosphate buffer, pH 6.85, 2.5 g $KH_2PO_4$ + 1.5 g $Na_2HPO_4 \times 2H_2O$ to 1 L) is delivered with an HPLC pump at a flow rate 1 mL/min to a size-exclusion column. The molecular weight calibration standard was comprised of polyethylene glycols (PEGs: 600–8000).

OCD and UVD calibration was based on potassium hydrogen phthalate. Customized software (SEC-OCD software, Young-in, Seoul, Korea) was used to quantify the organic carbon concentration.

### 2.4. Nematodes Analysis during GDM Filtration

The nematodes inhabited in the fouling layer were analyzed at the operation cycles of 74, 78, 85, and 93 cycles for PES-UF and 89, 93, 100, and 108 cycles for PVDF-UF and PTFE-MF. The 2 cm × 13.3 cm or 2 cm × 11.2 cm of biofilm were scraped from the membrane and resuspended in 1 mL of the water from the container. Individual nematodes were mounted onto slides using a platinum pick wire under a dissecting stereo microscope (Olympus SZ51, Tokyo, Japan) and observed under a bright-field microscope with differential interference contrast filter (Zeiss AxioImager A2, Jena, Germany), from which the number of nematodes was counted. The nematodes species were identified using 16S rRNA gene sequence analysis. Single nematodes were transferred to a 1 mL tube containing 10 μL of M9 buffer and 70 μL of phosphate buffer saline (PBS) and homogenized for 1 min using a grinder (Gingko Bioscience Company, Beijing, China) and then added with 100 μL of lysis buffer and 20 μL of proteinase K (QIAGEN Ltd., Crawley, UK). The tubes were incubated at 56 °C for 1 h and treated according to the manufacturer's protocol for the QIAamp DNA mini kit (QIAGEN Ltd., Crawley, UK). The 50 μL of eluted DNA was stored at −20 °C. The 10 μL of DNA suspension was added to the PCR mixture including primers SSU18A (5′-AAAGATTAAGCCATGCATG-3′) and SSU26R (5′-CATTCTTGGCAAATGC TTTCG-3′). The following PCR profile was used: 95 °C for 5 min; 40 × (94 °C for 60 s, 55 °C for 90 s, 72 °C for 120 s) and 72 °C for 10 min. The PCR products were confirmed by 1% agarose gel electrophoresis and stained with ethidium bromide and purified with the EZ spin column DNA gel extraction kit (Shanghai Sangon Biological Engineering Technology & Services Co. Ltd., Shanghai, China). Purified products were sequenced in both directions using the BigDye Terminator v3.1 Cycle Sequencing Kit (Applied Biosystems) and an AB PRISM 3730 (Applied Biosystems, Foster City, CA, USA) automatic sequencer. The evolutionary analyses were conducted in MEGA6 [31].

### 2.5. Statistical Analysis

Sigmaplot 14.0 was used to compare the statistical differences with $p < 0.05$ as the significance cut-off.

## 3. Results and Discussion

### 3.1. Surface Properties of UF and MF Membranes

Table 1 summarizes the surface properties of the two ultrafiltration (UF) membranes (PES-UF and PVDF-UF) and one microfiltration membrane (PTFE-MF) used in this study. The pore size of the membranes was provided by the manufacturers. The values of zeta potential were −35 mV for PES-UF and −37.4 mV for PVDF-UF. The zeta potential of the PTFE-MF membrane could not be determined, but the value reported in the literature is −22 mV [32]. The contact angles of the membranes were less than 50°, indicating that these membranes are all hydrophilic. In terms of roughness, the PES-UF membrane exhibited a smooth topology (average surface roughness (Ra) = 63.6 ± 15.4 nm). PVDF-UF and PTFE-MF membranes indicated a relatively rougher surface (188.8 ± 61.5 and 247.8 ± 54.3 nm, respectively) with a height variation of hundreds of nanometers.

**Table 1.** Surface properties of the membranes.

| Membranes | MWCO [a] (Pore Size *) | Zeta-Potential (mV) | Contact Angle (°) | Roughness (Ra, nm) [b] | Permeability (LMH/bar) |
|---|---|---|---|---|---|
| PES-UF | 100 kDa | −35.0 ± 3.7 | 47.5 ± 1.1 | 63.6 ± 15.4 | 120 ± 2 |
| PVDF-UF | 100 kDa | −37.4 ± 2.4 | 36.9 ± 0.6 | 188.8 ± 61.5 | 921 ± 21 |
| PTFE-MF | 0.3 μm * | −22 [c] | 37.0 ± 0.3 | 247.8 ± 54.3 | 8923 ± 104 |

[a] MWCO indicates nominal molecular weight cut-off. [b] Ra is the average surface roughness. [c] Zhang et al. (2020) [32].
* Pore size indicates mean pore size of the membrane surface.

### 3.2. Flux Behavior during GDM Filtration

Figure 2 shows the selected permeate flux behaviors of PES-UF, PVDF-UF, and PTFE-MF membranes at the beginning (cycle 1), middle (cycle 49 for PES-UF/cycle 65 for PVDF-UF and PTFE-MF), and end (cycle 96 for PES-UF/cycle 110 for PVDF-UF and PTFE-MF) of the GDM filtration cycle. The initial permeate fluxes were 53 LMH for PES-UF, 60 LMH for PVDF-UF, and 147 LMH for PTFE-MF (cycle 1). The different initial permeate fluxes were expected because of their clean membrane permeability, which was 120 LMH/bar for PES-UF, 921 LMH/bar for PVDF-UF, and 8923 LMH/bar for PTFE-MF. The higher flux of PTFE-MF compared to the other two UF membranes can be understood by its relatively larger pore size. Permeate fluxes dropped sharply to approximately 7 LMH due to both membrane fouling and the decreased hydrostatic pressure from 0.01 bar to 0.075 bar. In the middle of the GDM operation period (cycle 49 for PES-UF and cycle 65 for PVDF-UF and PTFE-MF), permeate fluxes were much lower compared to those at cycle 1. At cycle 65, the initial fluxes for PVDF-UF and PTFE-MF membranes were 11.8 and 11.1 LMH and decreased to 4.3 and 5.1 LMH, while the flux for PES-UF decreased from 3.3 LMH to 1.0 LMH at cycle 49. Similar trends were observed at the end of the operation (cycle 96 for PES-UF and cycle 110 for PVDF-UF and PTFE-MF). These results show that only PTFE-MF showed higher permeate flux at the beginning, and relatively higher fluxes were observed for PTFE-MF and PVDF-UF than PES-UF during the entire GDM operation period.

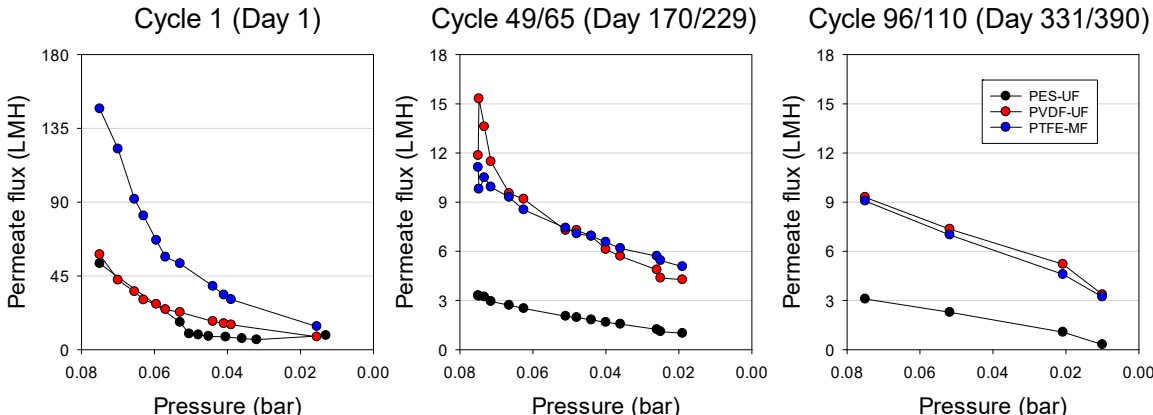

**Figure 2.** Permeate flux development depending on pressure drop from 0.075 bar to 0.01 bar during gravity-driven membrane (GDM) filtration with different membranes. The permeate flux was identified based on the operation cycle and time: cycle 1 (day 1), cycle 49 (day 170) = PES-UF and cycle 65 (day 229) = PVDF-UF/PTFE-MF, and cycle 96 (day 331) = PES-UF and cycle 110 (day 390) = PVDF-UF/PTFE-MF.

The mean permeability of each cycle could be calculated from the average of the permeate fluxes divided by the corresponding hydraulic pressure during GDM filtration (Figure 3). The initial mean permeability (cycle 1) was 312 ± 202 LMH/bar for PES-UF, 491 ± 111 LMH/bar for PVDF-UF, and 1172 ± 372 LMH/bar for the PTFE-MF membrane. The mean permeability for PES-UF gradually decreased and stabilized, while those for both PVDF-UF and PTFE-MF fluctuated. After cycle 5, a relatively higher mean stabilized permeability was observed for PVDF-UF and PTFE-MF, 192.9 ± 170.4 and 180.6 ± 120.9 LMH/bar, respectively, while PES-UF showed a much lower mean stabilized permeability at 46.1 ± 31.5 LMH/bar. These results can be explained by the effect of membrane surface properties on the fouling layer characteristics. Particularly, membrane surface roughness has a strong relationship with a mean stabilized permeability. Membrane surface roughness can affect the attachment and accumulation of fouling layers or predation and movement of eukaryotes within the fouling layer. These impacts can change the fouling layer structure. Therefore, the presence of a biologically active fouling layer on PVDF-UF and PTFE-MF membrane surfaces increased the permeability (or permeate flux) but also caused the highly fluctuating permeability. The membranes with higher roughness (i.e., PTFE-MF and PVDF-UF) showed higher fluxes than the smooth membrane

(i.e., PES-UF) in our GDM operation system. This was contrary to the typical flux behaviors of conventional MF/UF systems where rough membranes show lower permeability due to more severe membrane fouling [33–37]. The reason for this phenomenon is further discussed in Section 3.6.

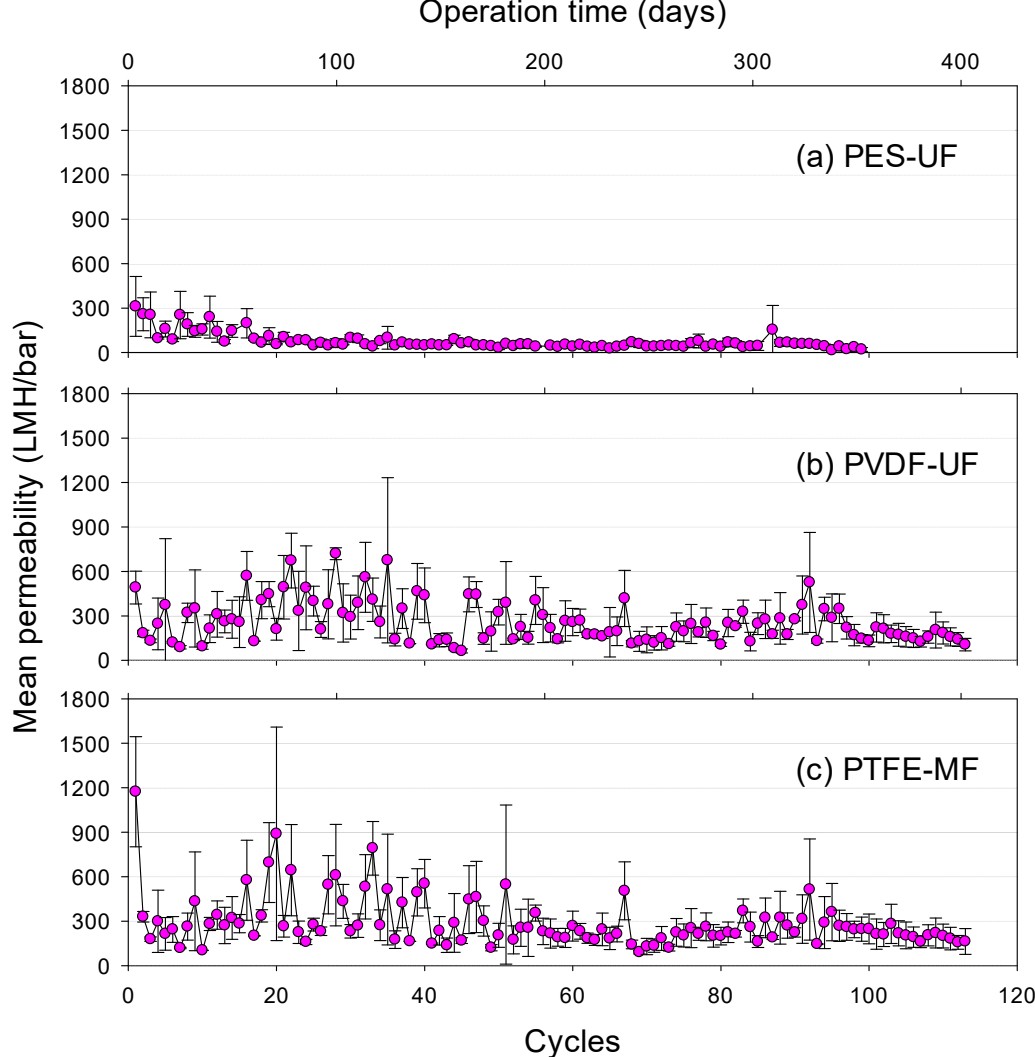

**Figure 3.** Mean permeability during GDM filtration using (**a**) PES-UF, (**b**) PVDF-UF, and (**c**) PTFE-MF membranes with an input of lake water as a function of operation cycles and time (mean permeability = average permeate flux divided by hydraulic pressure at each cycle, error bar represents the standard deviation during several measurements at each cycle).

### 3.3. Fouling Layer Characteristics

The morphology of the fouling layer on membrane surfaces was monitored at macro- and micro-scales using a digital camera (top-view images) and a scanning electron microscope (SEM). Figure 4 shows the top view images of the fouling layers at different operation cycles of GDM filtration. During the operation, the fouling layer gradually covered the membrane surfaces and became darker and heterogeneous. It was clear that the fouling of PVDF-UF and PTFE-MF was much more severe than that of PES-UF. The morphology of the PES-UF fouling layer barely changed with increasing operation cycles. The fouling layer on PVDF-UF and PTFE-MF showed dark brown mounds and heterogeneous structures. Coverage of the PVDF-UF and PTFE-MF membrane increased from 32–46% on day 23 to ~100% at the end of the operation, while that of PES-UF remained significantly lower (1–24%).

| Operation Cycle (days) | 1(1) | 11 (47) | 19 (80) | 33 (131) | 41 (160) | 48 (184) | 54 (206) | 64 (232) | 71 (250) | 89 (329) |
|---|---|---|---|---|---|---|---|---|---|---|
| PES-UF | | | | | | | | | | |

| Operation Cycle (days) | 1 (1) | 7 (23) | 16 (55) | 29 (109) | 36 (142) | 49 (193) | 64 (246) | 74 (279) | 86 (312) | 101 (391) |
|---|---|---|---|---|---|---|---|---|---|---|
| PVDF-UF | | | | | | | | | | |
| PTFE-MF | | | | | | | | | | |

**Figure 4.** Top-view images of the fouling layer on the membrane surface as a function of operation cycles and days.

Figure 5 shows top-view and cross-sectional SEM images of PES-UF, PVDF-UF, and PTFE-MF at the end of the filtration. The fouling layers developed on PVDF-UF and PTFE-MF had highly thick and complex structures (Figure 5e,f), while PES-UF showed a less developed fouling layer (Figure 5b,c). The thicknesses of the fouling layer, obtained from cross-sectional SEM images, were $1.1 \pm 0.5$ μm (PES-UF), $72.6 \pm 24.5$ μm (PVDF-UF), and $73.8 \pm 26.1$ μm (PTFE-MF). The green arrows indicate the interface between the membrane and the fouling layer. Unlike the heterogeneous and thick fouling layers on the PVDF-UF and PTFE-MF membranes, PES-UF showed a homogeneous and thin fouling layer. Although PVDF-UF and PTFE-MF showed increased thickness and coverage of the fouling layer, the stabilized permeability values were much higher compared to PES-UF (Figure 3), indicating that the thicker fouling layers of PVDF-UF and PTFE-MF were loosely formed and enabled higher permeate flux.

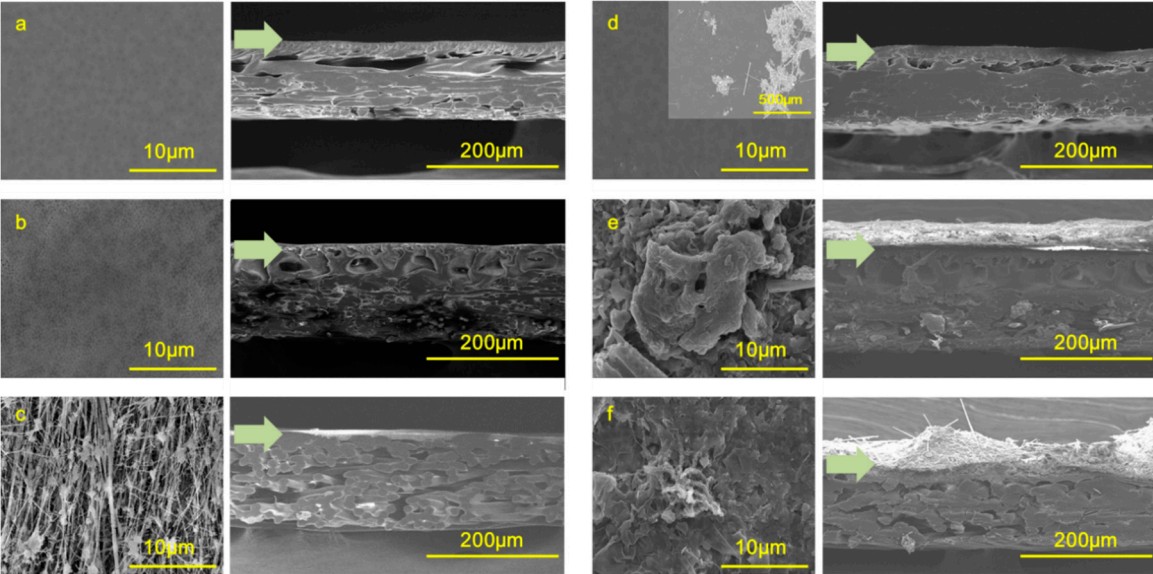

**Figure 5.** Top-view and cross-sectional SEM images of the membrane surfaces: (**a**,**d**) PES-UF, (**b**,**e**) PVDF-UF, and (**c**,**f**) PTFE-MF. Images (**a**–**c**) show clean membranes, and (**d**–**f**) are images of the fouled membranes taken at the end of the GDM filtration operation. The green arrows indicate the boundary between the membrane surface and the fouling layer.

Figure 6 shows the results of Student's *t*-test among the membranes in terms of mean permeability, roughness, and biomass concentration. ATP concentrations of the fouling layers were measured as the indicator of active biomass in the fouling layer. Measured biomass concentrations were $0.16 \pm 0.01$ gATP/m$^2$ for PES-UF, $1.13 \pm 0.63$ gATP/m$^2$ for PVDF-UF, and $0.96 \pm 0.63$ gATP/m$^2$ for PTFE-MF. Note that PES-UF had a much lower ATP concentration than PVDF-UF and PTFE-MF; the amount of active biomass of the fouling layer was 70% higher in PVDF-UF and PTFE-MF than in PES-UF. The fouling layer of PES-UF showed low microbial activity and may produce a lower amount of soluble microbial products [38]. More biomass accumulated onto the rougher surfaces of the PVDF-UF and PTFE-MF membranes.

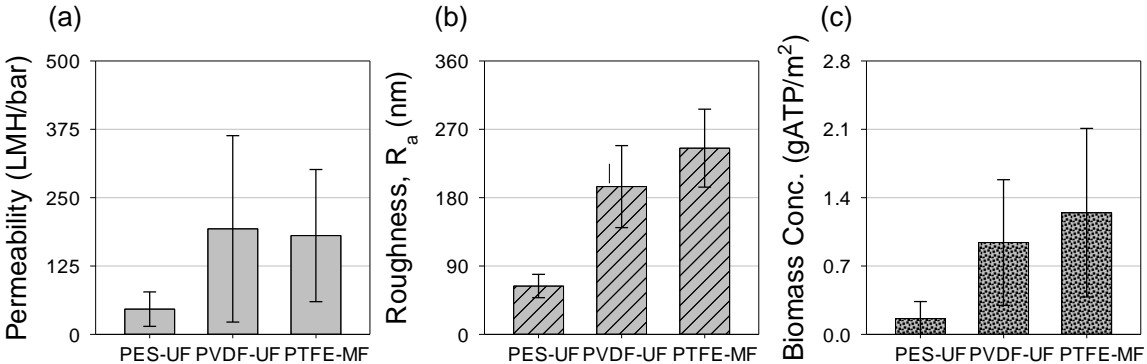

**Figure 6.** Measured membrane (**a**) mean stabilized permeability, (**b**) surface roughness of the clean membrane, and (**c**) ATP (adenosine triphosphate) content of the fouling layers. The mean permeability and ATP content of the fouling layer were obtained at operation cycles 81, 83, 85, and 93 for PES-UF, and 98, 100, 102, and 110 for PVDF-UF and PTFE-MF.

There was no significant difference in the stabilized permeability, membrane roughness, or biomass concentration between PVDF-UF and PTFE-MF ($p > 0.05$), whereas PES-UF showed statistically significant differences compared to both PVDF-UF and PTFE-MF ($p < 0.05$). These results indicate that the membrane roughness is closely related to the formation of the fouling layer, and the rougher membranes (PVDF-UF and PTFE-MF) led to more accumulation of particles and clogging of the valley than the more smooth membrane (PES-UF) [34,36]. It appears that different pore size and clean membrane permeability have less impact on the stabilized permeability, while the roughness of the clean membrane controls the permeability development.

### 3.4. Water Quality Changes during GDM Filtration

Table 2 represents the characteristics of dissolved organic matter (DOM) fractions in feed and permeate water during GDM filtration at operation cycles 16, 62, 73, and 81 for PES-UF, and cycles 31, 77, 88, and 96 for PVDF-UF and PTFE-MF. The DOM fractions could be quantified as biopolymer (>10 kDa), humic acids and building blocks (0.5–1 kDa), low molecular weight (LMW) acids (<0.5 kDa), and LMW neutrals (<0.5 kDa), based on the size exclusion chromatography–organic carbon detector (SEC–OCD) fractionation scheme [14,29,39]. As shown in Table 2, there were no significant changes of small molecules (<1 kDa), such as DOC concentration, humic acids, building blocks, and LMW neutrals, between feed water and permeates from three membranes. The concentration of biopolymer decreased after the GDM filtration; the removal rates of biopolymer were 16.3% for the PES-UF membrane, 18.8% for the PVDF-UF membrane, and 20.8% for the PTFE-MF membrane. The removal of a biopolymer indicated that this organic fraction accumulated on the membrane surface affecting the fouling layer characteristics. Additionally, higher concentrations of LMW acids were observed in the permeate, compared to the feed water. LMW acids can be affected by the release and degradation of organic matter from the biological fouling layer [40]. Moreover, its variation represented that biofouling or microbial activity plays a significant role in fouling layer characteristics.

**Table 2.** Dissolved organic matter (DOM) characteristics of feed and permeate water during GDM filtration.

| PES-UF/PVDF-UF and PTFE-MF Cycles (Days) | Parameter | Feed Water | PES-UF Permeate | PVDF-UF Permeate | PTFE-MF Permeate |
|---|---|---|---|---|---|
| 16/31 (56/115) | DOC (mg/L) | 2.8 | 2.5 | 2.5 | 2.5 |
| 62/77 (216/275) | | 2.7 | 2.7 | 2.7 | 2.7 |
| 73/88 (247/306) | | 3.2 | 3.2 | 3.1 | 3.1 |
| 81/96 (285/344) | | 3.6 | 3.6 | 3.6 | 3.6 |
| 16/31 (56/115) | Biopolymer (mg/L) | 0.8 | 0.03 | 0.1 | 0.2 |
| 62/77 (216/275) | | 0.4 | 0.02 | 0.1 | 0.1 |
| 73/88 (247/306) | | 0.5 | 0.04 | 0.05 | 0.1 |
| 81/96 (285/344) | | 0.7 | 0.3 | 0.2 | 0.1 |
| 16/31 (56/115) | Humic acids (mg/L) | 0.9 | 1.1 | 1.2 | 1.1 |
| 62/77 (216/275) | | 1.1 | 1.2 | 1.2 | 1.2 |
| 73/88 (247/306) | | 1.3 | 1.0 | 1.5 | 1.4 |
| 81/96 (285/344) | | 1.7 | 1.8 | 1.6 | 1.3 |
| 16/31 (56/115) | Building blocks (mg/L) | 0.3 | 0.4 | 0.4 | 0.4 |
| 62/77 (216/275) | | 0.3 | 0.4 | 0.4 | 0.4 |
| 73/88 (247/306) | | 0.4 | 0.4 | 0.5 | 0.5 |
| 81/96 (285/344) | | 0.4 | 0.4 | 0.5 | 0.4 |
| 16/31 (56/115) | LMW neutrals (mg/L) | 0.4 | 0.5 | 0.5 | 0.4 |
| 62/77 (216/275) | | 0.4 | 0.5 | 0.5 | 0.5 |
| 73/88 (247/306) | | 0.4 | 0.5 | 0.5 | 0.5 |
| 81/96 (285/344) | | 0.5 | 0.5 | 0.7 | 0.5 |
| 16/31 (56/115) | LMW acids (mg/L) | 0.4 | 0.5 | 0.4 | 0.4 |
| 62/77 (216/275) | | 0.4 | 0.7 | 0.5 | 0.5 |
| 73/88 (247/306) | | 0.6 | 1.2 | 0.6 | 0.6 |
| 81/96 (285/344) | | 0.3 | 0.6 | 0.7 | 1.2 |

### 3.5. Nematodes

The number and species of nematodes in the fouling layer on the membranes were characterized using a bright-field microscope and 16S rRNA sequencing, and the results are shown in Table 3 and Figure 7. Data were obtained from operation cycles 74, 78, 85, and 93 for PES-UF and cycles 89, 93, 100, and 108 for PVDF-UF and PTFE-MF. Table 3 summarizes the species of nematode observed in the membrane fouling layer. Six nematode species were identified, which were *Prismatolaimus intermedius*, *Rhabdolaimus aquaticus*, *Rhabdolaimus Terrestris*, *Rhabdolaimus* cf. *terrestris*, *Eumonhystera* cf. *vurgaris* and *Eumonhystera* cf. *hungarica*. The *Rhabdolaimus* and *Eumonhystera* genera were dominant within the fouling layer (69% and 23%, respectively). The length of the nematodes ranged from 200 to 500 μm.

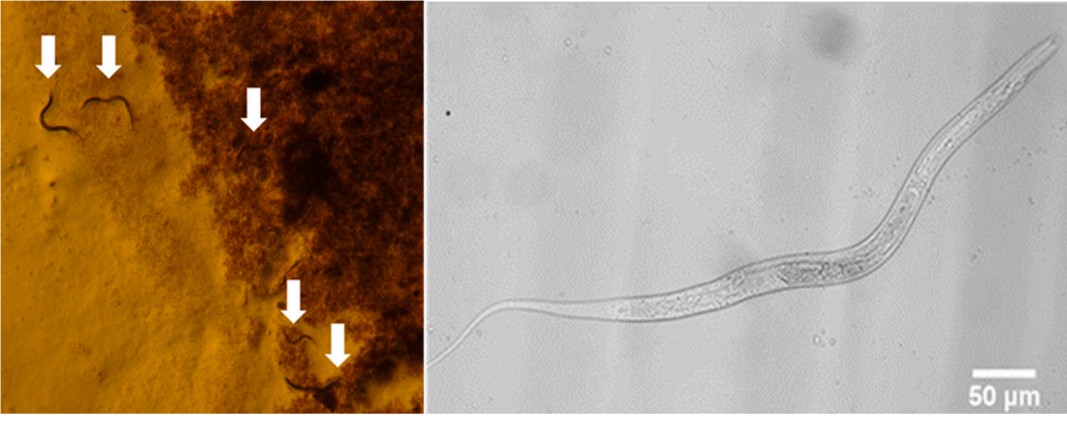

**Figure 7.** Microscopic images of nematodes on the PTFE-MF membrane surface during GDM filtration.

**Table 3.** Nematode species observed on the membranes after GDM filtration.

| Membrane | Closest Match (Accession Number) | Identity Score (% Similarity) |
|---|---|---|
| PES-UF | N.D. [a] | - |
| PVDF-UF | *Eumonhystera* cf. *hungarica 1JH-2014* (KJ636237.1) | 98 |
|  | *Monhystera* cf. *paludicola JH-2014* (KJ636247.1) | 98 |
|  | *Plectus minimus* (KC206040.1) | 99 |
| PTFE-MF | *Monhystera* cf. *paludicola JH-2014* (KJ636247.1) | 95 |
|  | *Plectus aquatilis* (GQ892827.1) | 100 |
|  | *Plectus minimus* (KC206040.1) | 99 |
|  | *Plectidae* sp. (AJ966508.1) | 100 |
|  | *Rhabdolaimus aquaticus* (AY284729.1) | 93 |
|  | *Rhabdolaimus terrestris* (AY284729.1) | 99 |
|  | *Rhabdolaimus* cf. *terrestris JH-2004* (KJ636250.1) | 96 |

[a] N.D. represents not detectable.

Figure 7 shows some representative images of nematodes observed on the PTFE-MF membrane. Based on the direct microscopic observations, higher numbers of nematodes were found in the fouling layers of PVDF-UF ($31 \pm 14$ count/cm$^2$) and PTFE-MF ($22 \pm 6$ count/cm$^2$), while nematodes were not observed on the surface of PES-UF. Many nematodes were observed on the rougher surface of PVDF-UF and PTFE-MF, with higher biomass concentrations compared to PES-UF. The activity of nematodes loosens the fouling layer and leads to a higher permeability with large fluctuations (Figure 3). Less biomass was accumulated onto the PES-UF, which has a smoother surface that nematodes preferred not to inhabit. The absence of nematodes in the fouling layer of PES-UF led to a gradual decrease in permeability without flux recovery.

*3.6. Relationship among Permeability, Number of Nematodes, and Biomass Concentration*

Figure 8 presents the relationships among stabilized permeability, the number of nematodes, and active biomass concentrations. As shown in Figure 8a, the number of nematodes increased at higher biomass concentrations on the membrane surface ($R^2 = 0.8$), indicating that biomass concentration in the fouling layer could be responsible for the higher number of nematodes. Abundant biomass could provide not only a favorable habitat for nematodes but also main food sources [10,41]. In Figure 8b, permeability increased with increasing biomass concentration in the fouling layer ($R^2 = 0.74$). This result was contrary to reports that the fouling layer deposited on membrane surfaces decreased permeability due to pore blocking and/or pore constriction [42–44]. However, the presence of eukaryotes (i.e., nematodes in this study) inhabiting the fouling layer increased the permeate flux by 50–170% during GDM filtration in a previous study [9]. Similarly, permeability was found to increase with the increasing number of nematodes in this study ($R^2 = 0.72$). Metazoan organisms (e.g., nematodes) can graze and feed on a wide range of food sources (bacteria, algae, organic detritus, and protozoa) within the fouling layer. Moreover, an improved population of nematodes results in a significant reduction of the basal layer [9]. The predation of eukaryotes could have decreased the density and thickness of the fouling layer, resulting in the formation of heterogeneous and porous fouling layer structures. Therefore, a rougher membrane surface appears to accumulate more biomass in the fouling layer, which is favorable for eukaryotes to inhabit, thereby enhancing permeability in GDM filtration.

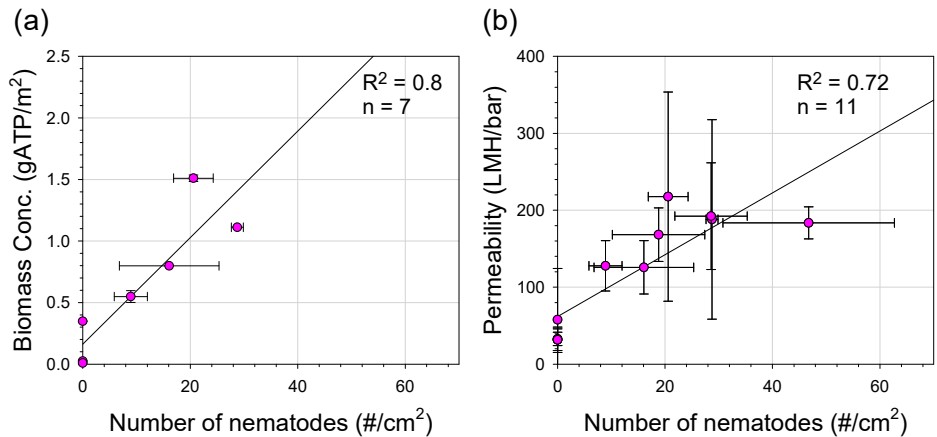

**Figure 8.** (**a**) Biomass concentration versus nematode number on the membrane surface, stabilized permeability versus biomass concentration on the membrane surface, and (**b**) stabilized permeability versus nematode number on the membrane surface. The stabilized permeability, biomass concentration, and nematode number were measured at operation cycles 74, 78, 81, 83, 85, and 93 for PES-UF and cycles 87, 95, 98, 100, 102, and 110 for PVDF-UF and PTFE-MF.

## 4. Conclusions

This study evaluated the permeability of submerged GDM filtration with an intermittent operation mode using an input of lake water for about one year. PES-UF, PVDF-UF, and PTFE-MF membranes with different surface properties were tested. The average permeability was determined to be 192.9 LMH/bar for PVDF-UF, 180.6 LMH/bar for PTFE-MF, and 46.1 LMH/bar for PES-UF. The different permeabilities among the tested membranes during GDM filtration were closely related to the membrane roughness, biomass concentration, and eukaryotic (nematodes) inhabitants. PVDF-UF and PTFE-MF, characterized by rougher membrane surfaces, accumulated more biomass in the fouling layers where the nematodes were more likely to be found. Due to the activity of nematodes, the fouling layers became more porous, resulting in increased permeability. PES-UF with its smoother membrane surface accumulated less biomass in which no nematodes were observed, resulting in low permeability. A good correlation between the number of nematodes and permeability was observed ($R^2 = 0.72$), providing important insights into the selection of the membrane type as a critical parameter affecting the performance of GDM filtration.

**Author Contributions:** D.L., S.S.C., and Y.L. designed the experiments and goals of the research study; D.L., Y.J.C., and Y.B. analyzed data and wrote the manuscript; S.S.C. and Y.L. are responsible for overall content of the research. All authors have read and agreed to the published version of the manuscript.

**Funding:** This work was supported by GIST Research Institute (GRI) grant funded by the GIST in 2020 and the Korea Institute of Energy Technology Evaluation and Planning (KETEP) and the Ministry of Trade, Industry & Energy (MOTIE) of the Republic of Korea (No. 20191510301170).

**Conflicts of Interest:** The authors declare no conflict of interest.

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
