# Peer review of "Relationships among Permeability, Membrane Roughness, and Eukaryote Inhabitation during Submerged Gravity-Driven Membrane (GDM) Filtration"

_applsci, doi:10.3390/app10228111_

Round 1
Reviewer 1 Report
The work deals with the treatment of a lake water by gravity-driven membrane to assess the effect of some surface properties on permeate flux, mean permeability and the behaviour related to the nematode population, as a representative metazoan species, on the fouling layer. The manuscript is well structured and results appear new and interesting for the water treatment by membranes. Thus, the manuscript may be considered for publication after the following revision:
- The scope of a treatment by membrane is the separation of any pollutant. However, in this work is found that many of the parameter followed during the treatment are not affected by the treatment. Only biopolymer is removed. This was the scope of the treatment? Please, explain what biopolymer mean and how its removal can affect the quality of the water.
- The authors should detail the chromatography analysis (mobile phase, flow rate and column dimensions) and also the identification and quantification of DOC, biopolymer, humic acids, building blocks, LMW neutrals and LMW acids? Please explain how those substances were measured and calibrated (was it used a software?). Furthermore, an explanation about the mean of “biopolymer” and “building blocks” is necessary.
- Authors can developed a greater discussion of the results obtained in first section: Is the Z-potential, hydrophilicity and roughness interesting for this process? How affect to the membrane separation? Are the values obtained similar to those found in the literature?
- The discussion of the Figure 2-3 may also be increased. Please, compare the results obtained with the characteristic of the membranes and the results found in the literature.
- In my humble opinion, the Student’s test represented in Figure 6 is not necessary to show the significant difference between the permeability, roughness and biomass concentration. I suggest to remove the lines 268-290 and prepare a discussion only relate to the biomass concentration and the results obtained from the analysis with ATP.
- Some missing specifications and other details in the methodology section:
- Please, specify the dimensions of the membranes (and the model of the membranes coming from Trisep Co. and Philos Co., Ltd. If available).
- Was the effective membrane area (0.0523 m2) the same for each one? Was it specified by the supplier or was it determined by the authors?
- Please, consider remove the second and third tank in Figure 1, since as it was designed the reader may understand that three tanks consecutively operating was studied. With only one tank, I suggest to design an arrow to specify the decrease of the water level from 0.075 bar to 0.01 bar at the beginning and the end of each cycle, respectively.
- The dimensions of the tank is missing.
- It would be interesting to explain how the membranes were disposed into the tank (was a model to support them used?).
- Editing mistakes to correct:
- In lines 119-123, the same sentence is almost repeated.
- Consider use (a), (b) and (c) In Figure 3 and axis titles between the graphs.
- The cross-sectional photo in Figure 5 for (a) and (c) is exactly the same microphotograph. Please check.
- In line 268, Figure 6 is follow by a dot “.” in the text.
- There are two sections numbered as 3.3. (“Fouling layer characteristics” and “Water quality changes during GDM filtration”). Consecutives sections should be renumbered.
- The caption of Figure 6 end with two dots “. .”.
- In Figure 8 was written “Number of nemaotdes”, nstead of “Number of nematodes”.
- According to the instruction for authos of the applied sciences, the following statements should be used in Author contributions: "Conceptualization, X.X. and Y.Y.; Methodology, X.X.; Software, X.X.; Validation, X.X., Y.Y. and Z.Z.; Formal Analysis, X.X.; Investigation, X.X.; Resources, X.X.; Data Curation, X.X.; Writing – Original Draft Preparation, X.X.; Writing – Review & Editing, X.X.; Visualization, X.X.; Supervision, X.X.; Project Administration, X.X.; Funding Acquisition, Y.Y.”.
- If there is not “Acknowledgments”, line 374 should be removed.
- The authors need to revise the references (some identified mistakes: reference 13 has the abbreviated name of the journal, reference 15 is incomplete, reference 20 is confused and unnecessary and reference 34 has the names of the authors in capital letters).
Reviewer 2 Report
The paper reports interesting results on the impact of surface roughness on the permeability of submerged membranes. Membrane permeability and changes in permeability were ascribed at the concentration of nematodes in the fouling layer. Such concentration depends on the surface roughness.
An experimental set-up was designed to simulate realistic filtration conditions and run over 110 cycles, thus obtaining a solid set of experimental data. However, I believe that a major revision is needed before considering the paper for publication.
- The authors find a relation among surface roughness, nematodes concentration and membrane permeability. In my opinion, the main fault of the paper is to do not discuss why nematodes and no other components of the fouled biomass is responsible for the increased permeability. . Does the fouled layer on PES-UF differ from the others only in the absence of nematodes? The authors must provide either experiment evidences or references to relevant literature, in order to demonstrate that nematodes are the main responsible for the permeability of the two membranes with high surface roughness.
Moreover, other comments are:
- What is the permeability of the membrane with no fouling? A value can be easily obtained by filtering demi or tap water for a short time.
- The abstract should be rewritten considering the following observation
What is the rationale in selecting these three membranes?
The authors write that “the observed higher permeability …. ….was thought to be related to the rougher surface.” However, permeability is mainly related to pore size (or MWCO) and those values should be reported in the abstract. Pore density (porosity or pore fraction) also plays a role and it can be reported in the abstract, if available.
Explain the meaning of “LMH” also in the abstract
- Figure 3. Please, correct PTFE-UF with PTFE-MF
- The statistical analysis and the meaning of the P values should be explained better in the main text and in the figure caption
Reviewer 3 Report
The authors of the present paper devote their work to study a new and
apparently promising membrane process, the Gravity-driven membrane
filtration. The technique is basically simple as it uses the pressure
due to a column of water to avoid the use of any pumping through the
membrane or any backflushing to avoid membrane fouling or at least to
restore eventually membrane performance. This method supplies a quite
low transmembrane pressure and, consequently, much smaller fluxes. The
main advantage of such method is , in my opinion, the economic advantage
of avoiding pumping, then energy consumption, to filtrate the batches.
This advantage must be balanced with the lower fluxes obtained which
make treating a gentle amount of feed liquid needs a big membrane area
and since membranes are the major investment cost in a membrane based
treatment plant, is not clear for me, if finally it will be competitive.
Another advantage relies in the fact that the biofilm developed onto the
membrane surface can control the permeability leading to a
pseudomembrane layer which at the end governs flux and retention.
Attending to the paper here presented this biofilm is more porous than
usual cakes appearing when pumping pressure is applied (which is
certainly reasonable but not novelty) and it is also controlled by the
presence of nematodes which contribute to maintain the porous structure
of such deposited layer. Certainly it seems quite difficult to maintain
the long term performance of a process which depends on the presence of
a gentle number of nematodes. Anyway the technique seems useful in some
cases where it is needed to use simple and not expensive filtration
systems to get enough quality filtered water. In that sense the aim of
the paper can be considered of interest.
Regarding the manuscript, it needs some extensive rewriting with many
paragraphs and explanations quite unclear. One of my major concern comes
from the fact that, reading the manuscript, seems author have used some
usual techniques to measure or characterize their results but they have
not clear what are they measuring. As a clear example, the measurement
of zeta potential. According to section 2.1 and table 1, authors have
measured the zeta potential of the three membranes used in their study.
This makes no sense, zeta potential of a membrane surface depends on the
electrostatic properties of the solution filtered through the membrane
and the interaction of the charges present in that solution with the own
membrane surface charges. Consequently it is erroneous to assign a
unique value of zeta potential for a membrane surface. It changes
depending on the ionic strength and pH of the solution and usually it
could change from positive to negative, passing through the so called
zero charge point. Moreover, the zeta potential must be measured through
streaming potential or electroviscous potential determinations ,
therefore electrokinetic processes involving transport of solutes
through or onto the membrane which lead to develop the electric double
layer in the membrane surface. In any case, it cannot be measured by
electrophoretic mobility determinations. This method, as apparently is
the one used by the ELS-8000 device, is the common way to determine the
zeta potential of moving particles in a solution. Are those particles
the polystyrene latex ones described in lines 100-101. IN such case, you
are measuring the zeta potential of the latex particles not that of the
membrane. This conceptual error, by itself, makes me to reject this
manuscript. Authors should have clear what they really measured and if
I’m wrong in my concerns, they should write the manuscript in a clearer
way to avoid such misunderstandings.
Some minor comments follow:
- Line 85: nominal MWCO, as given by manufacturer, not measured
- Line 86: …with a mean pore size of ….
- Line 89: it is not clear if Amicon cell was used only for measuring
initial water permeability or later as membrane cell during submerged
GDM. Amicon cell have different dimensions of 200 mm × 140 mm × 49 mm).
- Lines 95-96; rewrite completely, now stands unclear
- Figure 2: values are presented for cycle 1, cycles 49/65 and cycles
96/110. Why these selection of cycles? And the value for cycles 46/95
what does it mean, the average fo the fluxes found during cycles 46 to
95? If so, why not averaging cycles 1 to 6, in a similar manner?
- Figure 3: pressure drop changes continuously as water level falls
in the system (gravity filtration). Then seems very complicated to
determine simultaneously flux and pressure, as both are simultaneously
changing. how did you managed?
- Figure 5: regarding the arrow, which is the membrane and which is
the biofilm? The last one seems indistinguishable in most cases.
- Figure 6: very unclear, explain what means the lines and pressure
values inserted on it
- Table 2: similarity to a previous comment I cannot see which is the
reason to select such cycles, since they are not equally spaced though
the time span of the whole experiment. are them the most representative?
- Figure 7: I wonder how do you manage to differentiate several kind
of nematodes from these figures. On the other side, Nematodes were just
counted one by one? or did you use ImageJ. And how?
- Figure 8: there is a typo in the x axis (nematodes). Also you
present 3 figures showing different dependences but one of these is
redundant by the transitive property (if A linearly depends from B and B
linearly depends from C, then A linearly depends from A, any other
possibility should be quite surprising).
Round 2
Reviewer 2 Report
The authors modified the manuscript according to the referees' comments, providing also new data. Therefore, I recommend the manuscript for publication in Applied Science.
Reviewer 3 Report
Two questions still unclear for me.
- Certainly zeta potential of the surface can be measured by the method proposed. But it is also clear, see ref. 22, that the method has lower accuracy than measuring it from streaming potential or streaming current method. Moreover what are you quoting is a single value of zeta potential corresponding to a very specific conditions of charge of the test particles and measuring solution. So these conditions must be clearly specified.
- It is still unclear for me how you can manage to distinguish individual nematodes (so similar apparently) and at the same time, counting a great number of them. It should be a very slow and uncertain way of working.